



# Recently fixed carbon fuels microbial activity several meters below the soil surface

Andrea Scheibe[1], Carlos A. Sierra[2,3], Marie Spohn[*1,4]

[1]Bayreuth Center of Ecology and Environmental Research (BayCEER), University of Bayreuth, Germany
[2]Department of Biogeochemical Processes, Max Planck Institute for Biogeochemistry, Jena, Germany
[3]Department of Ecology, Swedish University of Agricultural Sciences, Uppsala, Sweden
[4]Department of Soil and Environment, Swedish University of Agricultural Sciences, Uppsala, Sweden

*Correspondence to*: Marie Spohn (marie.spohn@slu.se)

**Abstract.** The deep soil, >1 meter, harbors a substantial share of the global microbial biomass. Currently, it is not known whether microbial activity several meters below the surface is fueled by recently fixed carbon or by old carbon that persisted in soil for several hundred years. Understanding the carbon source of microbial activity in deep soil is important to identify the drivers of biotic processes in the critical zone. Therefore, we explored carbon cycling in soils in three climate zones (arid, mediterranean, and humid) of the Coastal Cordillera of Chile down to a depth of six meters, using carbon isotopes. Specifically, we determined the $^{13}C:^{12}C$ ratio ($\delta^{13}C$) of soil and roots, and the $^{14}C:^{12}C$ ratio ($\Delta^{14}C$) of soil and $CO_2$ respired by microorganisms. We found that the $\Delta^{14}C$ of the respired $CO_2$-C was higher than of the soil organic carbon in all soils (except for two topsoils). Further, we found that the $\delta^{13}C$ of the soil organic carbon changed only in the upper decimeters (by less than 6 ‰). Our results show that microbial activity several meters below the soil surface is mostly fueled by recently fixed carbon that is on average much younger than the total soil organic carbon present in the respective soil depth increments, in all three climate zones. Further, our results indicate that strong microbial decomposition of the soil organic matter only occurs in the upper decimeters of the soils, which is likely due to stabilization of organic carbon in the deep soil. In conclusion, our results demonstrate that microbial processes in the deep soil several meters below the surface are closely tied to primary production aboveground.



## 1. Introduction

The deep soil (>1 meter) harbors not only a large part of the global soil microbial biomass (Pedersen, 1997; Krumholz, 2000; Akob and Küsel, 2011), but also large amounts of organic carbon (C) (Rumpel and Kögel-Knabner, 2011; Marin-Spiotta et al., 2014; Jackson et al., 2017; Moreland et al., 2021; Marín-Spiotta and Hobley, 2022). In topsoils (<1 m), microorganisms have been shown to rely primarily on young organic C, tying microbial activity belowground closely to photosynthetic activity aboveground (Trumbore, 2000; van Hees, 2005; Högberg and Read, 2006; Högberg et al., 2008). Whether this relationship holds for deep soils (> 1 m) has hardly been investigated yet, which limits the understanding of biotic processes, such as biogenic weathering, in the deep critical zone.

The age of C metabolized by microorganisms can be estimated by measuring the $^{14}$C signature of the C emitted by the soil microbial biomass in the form of $CO_2$ (Trumbore, 2000). To our knowledge, and according to the International Soil Radiocarbon Database (Lawrence et al., 2020), the $^{14}$C signature of $CO_2$ respired by microorganisms in soil below 1.0 m has been measured only in permafrost soils (Dutta et al., 2006; Lee et al., 2012; Gentsch et al., 2018), showing that microbial respiration during winter relies strongly on C that has persisted in these soils for centuries (Dutta et al., 2006; Lee et al., 2012; Gentsch et al., 2018; Pedron et al., 2022). For non-permafrost soils, there are no data on the $\Delta^{14}$C of $CO_2$ respired by microorganisms for depth increments below 90-100 cm (Nagy et al., 2018; Lawrence et al., 2020). Although the $\Delta^{14}$C of total soil $CO_2$ at greater depths has been reported (Trumbore et al., 1995; Fierer et al., 2005), the total $CO_2$ is largely composed of $CO_2$ respired by roots and, therefore, masks the information about soil C cycling by microbial processes.

To test the hypothesis that microbial activity in deep, non-permafrost soil is driven by young organic C, we explored C cycling in soils of the Coastal Cordillera of Chile using isotopes. The bedrock in this region is extraordinarily deeply weathered (Vázquez et al., 2016; Hayes et al., 2020; Krone et al., 2021), possibly because of the $CO_2$ and organic acids produced by soil microorganisms (Berner, 1997; Berg and Banwart, 2000; Hoffland et al., 2004; Finlay et al., 2020; Uroz et al., 2022). We studied soils in the arid, mediterranean and humid climate zone that correspond to the vegetation zones arid shrubland, sclerophyllous forest, and humid temperate forest, respectively (Fig. 1). For this purpose, three to four soil profiles per site were dug down to a depth of 6.0 m (humid and mediterranean site) or 2.0 m (arid site). Soil and root samples were collected at regular intervals. To determine the age of C metabolized by microorganisms, we determined the $\Delta^{14}$C of the C that was emitted by soil microorganisms in the form of $CO_2$. Further, we determined the $\Delta^{14}$C of total organic carbon (TOC) to estimate its age, and we also measured its concentration. To quantify the extent of microbial decomposition of organic matter in soil, we determined the $\delta^{13}$C of TOC and roots. Finally, in order to quantify microbial biomass and activity, we extracted and quantified DNA from soil, and determined the soil-microbial respiration rate.



## 2. Material and Methods

### 2.1 Study sites

All three study sites are located along a precipitation gradient in the Coastal Cordillera of Chile between 30°S and 38°S (Fig. 1; Table 1). Mean annual precipitation is about 87, 436 and 1084 mm at the arid, mediterranean and humid site, respectively. The arid site is located in the private reserve Santa Gracia (Table 1). The vegetation is classified as arid shrubland. The vegetation is sparse (10 % cover) and dominated by *Proustia cuneifolia*, *Cordia decandra* and *Adesmia spp.* (Spohn and Holzheu, 2021). The mediterranean site is located ~4.2 km south of the national park La Campana, in a water protection area. The soils are covered by a low Sclerophylous forest. The humid site is located at the eastern border of the national park Nahuelbuta. The soils are covered by a mixed humid, temperate forest with coniferous and deciduous tree species. Further information about climate, geology, pedogenesis, and vegetation are provided in Bernhard et al. (2018), Oeser et al. (2018), as well as Brucker and Spohn (2019).

### 2.2 Soils and sampling

The soils at all three sites have formed from plutonic bedrocks of similar granitoid lithology (Oeser et al., 2018). They were not affected by glaciation during the last glacial maximum (~19,000–23,000 years ago, Hulton et al., 2002), and have not received volcanic ashes (Oeser et al., 2018). The soils of the arid and mediterranean site are classified as Cambisols, and the soils at the humid site are classified as Cambisols and Umbrisols (IUSS Working Group WRB, 2015).

The soil sampling was conducted at the arid site in March 2019 and at the mediterranean and humid site in March 2020. For the soil sampling, three soil profiles down to a depth of 200 cm were excavated on each site in an area of 1 km$^2$. One of them was further extended to a depth of 600 cm at the mediterranean and humid site. Furthermore, at the humid site, an additionally fourth profile was established down to 400 cm soil depth. The soil sampling was conducted the day after excavation from the bottom to the top of the profiles in the following soil increments: from 600 cm to 200 cm in 50 cm increments and from 200 cm to 20 cm in 20 cm increments. For the uppermost 20 cm, the following soil increments were sampled: 0–5, 5–10, and 10–20 cm. A subsample of the organic layer on top of each profile was collected at the mediterranean and humid site (there is no organic layer at the arid site). In addition, root samples were taken to complement the roots collected with the soil.

The percentage of fine soil (< 2 mm) is smaller at the arid site than at the two other sites throughout all soil depth increments (Table S1). The percentage of fine soil (< 2 mm) strongly decreases with increasing soil depth at the arid site and less strongly at the humid site (Table S1). At the arid site, the soil texture of the fine soil fraction (< 2 mm), averaged over all profiles, is loamy sand in the uppermost 80 cm, silty sand from 80 to 100 cm soil depth and pure sand below 100 cm soil depth. At the mediterranean site, the soil texture in the uppermost 160 cm is medium to slightly loamy sand and below 160 cm soil depth pure sand. The soil texture at the humid site is loamy. All soils are free of carbonates.



### 2.3 Sample preparation and soil physical analyses

All field moist soil samples were sieved (Ø 2 mm) and roots and other debris removed by hand. The weights of both fine soil fraction ($sf_{<2mm}$) and gravel or stone fraction ($sf_{>2mm}$) were determined for each soil increment. After subtracting the soil water content (SWC), the proportion of the fine soil fraction ($sf_{<2mm}$) per mass of the total sample was calculated. Subsamples of the sieved soil (< 2 mm) were either immediately dried at 60°C (for chemical and isotope analyses), or stored at 5°C (for incubation experiments), or frozen at -20°C (for DNA analyses). The subsample of the organic layer was dried at 40°C (for chemical and isotope analyses). For each soil increment, the SWC of the field moist soil was determined gravimetrically. The soil water holding capacity (WHC) was determined for eight soil increments down to 600 cm (0–5, 5–10, 10–20, 40–60, 100–120, 180–200, 350–400 and 550–600 cm), in order to facilitate the adjustment of comparable SWCs in the incubation experiment. The WHC for the other soil increments were obtained by extrapolation between measured soil increments. For the determination of the WHC, field-moist soil samples were oversaturated with water for 24 h, then left on a sand bath for 24 h. Afterwards, the samples were weight before and after drying at 105°C for 24 h.

For the mediterranean and humid site, fine roots (Ø < 2 mm) per individual soil increment were collected during the sieving process. At the arid site, fine roots were not present in all soil increments, and therefore roots of different increments were combined to a composite sample to acquire enough biomass for the isotope analyses. All root samples were washed and subsequently dried at 60°C.

### 2.4 Chemical and stable isotope analyses

Subsamples of the organic layer, dried fine soil (< 2 mm) and root samples were milled and analyzed for their carbon (C) concentration as well as stable carbon isotope ratio ($\delta^{13}C$) using an element analyser (NA 1108, CE Instruments, Mailand, Italy) coupled to an isotope-ratio-mass spectrometer (delta S, Finnigan MAT, Bremen, Germany) via a ConFlo III interface (Finnigan MAT, Bremen, Germany). Subsequently, the soil total organic C (TOC) concentrations were re-calculated per total soil mass ($sf_{<2mm} + sf_{>2mm}$).

### 2.5 Soil respiration rates

To determine the soil respiration rate for each soil depth increment, a subsample of the fine soil (< 2 mm) was incubated in a 350 ml incubation jar with septum. The amount of soil (dry weight equivalent) used for the incubation differed between soil depth increments: 20 g (mediterranean and humid) or 30 g (arid) for 0–20 cm, 40 g for 20–100 cm, 60 g for 100–200 cm, 80 g for 200–400 cm and 100 g for 400–600 cm. After adjusting the SWC to 60 % WHC, the incubation jars were closed air tight and the samples pre-incubated at 20°C in the dark for 3 (arid) or 4 (mediterranean and humid) days. Subsequent to the pre-incubation period, the incubation jars were aerated for several min, closed again, and the $CO_2$ concentration within the incubation jars was immediately analysed (time point T0). Afterwards, the samples were incubated at 20°C in the dark for 6





weeks. During the whole incubation period, the $CO_2$ concentration in the incubation jars was measured weekly (and additionally for mediterranean and humid site at day 3). For this purpose, a gas sample of 50 µl or 100 µl was collected from the headspace of each incubation jar using a syringe and immediately measured at a gas chromatograph (SRI 8610C, SRI Instruments, USA). At the time point of gas injection, air temperature, air pressure and inner pressure of the incubation jar was noted for each sample at each measurement time point. The measured $CO_2$ concentration in ppm were converted into $CO_2$-C by taking into account the sum of air pressure and inner pressure of the incubation jar (p), the molar mass of C ($M_C$ = 12.01 g mol$^{-1}$), the air temperature (T), the universal gas constant (R = 8.314 [(kg *m$^2$) (s$^{2*}$ mol * K)$^{-1}$]), the conversion of [g] into [mg] and the ideal gas law using the following Eq. (1).

(1)

$$CO_2 - C \left[\frac{mg}{m^3}\right] = \frac{p \left[\frac{kg}{(m * s^2)}\right] * M_C \left[\frac{g}{mol}\right] * CO_2 \ [ppm]}{R \left[\frac{(kg * m^2)}{(s^2 * mol * K)}\right] * T \ [K]} * 1000$$

The soil respiration rate was calculated across the linear increase in the $CO_2$ concentration over the 6-weeks incubation period using a linear regression. Afterwards, the soil respiration rate per mass of the total sample [mg $CO_2$-C m$^{-3}$ d$^{-1}$] was calculated for the volume of headspace ($Vol_{HS}$), the amount of dry soil ($soil_{dw}$) within each incubation jar, considering the proportion of the fine soil fraction ($sf_{<2mm}$) in the total soil ($sf_{>2mm}$ + $sf_{<2mm}$), and including the conversion of [mg] into [µg] using the following Eq (2).

(2)

$$CO_2 - C \left[\frac{\mu g}{g * d}\right] = \frac{CO_2 - C \left[\frac{mg}{m^3 * d}\right] * Vol_{HS} \ [m^3] * \left(\frac{sf_{<2mm} \ [g]}{sf_{>2mm} \ [g] + sf_{<2mm} \ [g]}\right)}{soil_{dw} \ [g]} * 1000$$

The $Vol_{HS}$ was calculated as the difference between the total volume of the incubation jar (350 ml) and the volume of soil at 60 % WHC.

### 2.6 Extraction and quantification of microbial DNA

The microbial DNA (M-DNA) was extracted from 400 mg of field moist soil (< 2 mm) using a kit (FastDNA Spin Kit for Soil, MP Biomedicals, Solon, USA) following the instructions of the producer. The volume of the extracted M-DNA eluate was determined and afterwards the M-DNA was quantified using a pico green assay (Quant-iT PicoGreen dsDNA Assay Kit, Invitrogen, Life Technologies Corporation, Eugene, OR, USA) measured at a fluorescence microplate reader (FLx800, BioTek, Winooski, VT, USA). By considering the proportion of the fine soil fraction ($sf_{<2mm}$) in the total soil ($sf_{<2mm}$ + $sf_{>2mm}$) and the dry-weight equivalent of the soil the amount of M-DNA per mass of the total sample was calculated.





### 2.7 Radiocarbon ($^{14}$C) ratio of respired $CO_2$-C and soil TOC

To determine the radiocarbon ($^{14}$C) ratio of respired $CO_2$-C, fine soil (< 2 mm) subsamples were incubated in 1050 ml
incubation jars with septum. For the arid study site, between 100–340 g soil (dry weight equivalent) was incubated for the
different soil increments of the three profiles. For the mediterranean and humid site, the amount of soil (dry weight equivalent)
used for the incubation differed between soil depth increments: 180 g (humid) or 250 g (mediterranean) for 0–10 cm, 240 g
(humid) or 280 g (mediterranean) for 10–200 cm, 260 g (humid) or 330 g (mediterranean) for 200–600 cm. Immediately after
adjusting the soil water content to 60 % WHC, the incubation jars were closed air-tight and rinsed with synthetic air
(hydrocarbon free, Riessner-Gase GmbH, Lichtenfels, Germany) for ~6 min. with a flow rate of ~700 cm$^3$ min$^{-1}$ to remove the
atmospheric $CO_2$ within the incubation jars. Afterwards the samples were incubated at 20°C in the dark. Gas samples from the
headspace of the incubation jars were collected for the arid site at six times during an incubation period of ~11 month and for
the mediterranean and humid site at four times during an incubation period of ~8 month. The $CO_2$ concentration of the gas
samples was determined using a gas chromatograph and the amount of $CO_2$-C within the headspace of the incubation jar was
calculated (as described previously in Eq. (1)). Once a minimum amount of ~1 mg $CO_2$-C within the headspace of the
incubation jar was reached, the gas in the headspace was transferred into a 400 ml vacuum container (miniature air sampling
canister, Restek, Bellefonte, USA) for storage until further processing. For samples with a minimum amount of only ~0.5 mg
$CO_2$-C within the headspace of the incubation jar, the $CO_2$ was directly extracted from the incubation jar.

The $^{14}$C analyses of the respired $CO_2$-C and the soil TOC were conducted using the accelerator mass spectrometry (AMS)
facility in Jena, Germany (Steinhof et al., 2004). From the storage container or directly from the incubation jar, the respired
$CO_2$ was transferred into a glass tube cooled by liquid nitrogen and containing an iron catalyst. The following reduction of
$CO_2$ to graphite was carried out in the presence of hydrogen ($H_2$) at 600°C. The resulting graphite-coated iron was pressed into
targets and measured for $^{14}$C using the AMS MICADAS (Ionplus, Dietikon, Switzerland). To determine the $^{14}$C ratio of soil
TOC samples, milled soil subsamples were combusted and graphitized following Steinhof et al. (2017). The radiocarbon ratio
is reported as $\Delta^{14}$C in per mille [‰], which is the fraction with respect to the standard isotope ratio (oxalic acid standard SRM-
4990C; Steinhof et al., 2017) including the normalization for $\delta^{13}$C (fractionation correction) and the correction for the decay
between 1950 and the measurement time (2020/21; Stuiver and Polach, 1977).

We refrain from presenting radiocarbon ages in years because in open systems, such as soils, radiocarbon is continuously
exchanged with the atmosphere and traditional radiocarbon dating cannot be done (Trumbore et al., 2016). A mean age estimate
using the available radiocarbon data would require fitting a compartmental model to the data, for which there is not enough
time-resolved data, for instance about C inputs that cover the relevant timescale (last 1000 years), which in consequence would
lead to a lack in degrees of freedom of the model.





## 3. Results

The $\Delta^{14}C$ of the TOC was lower in soils of the arid site than in soils of the two other, less arid sites, across all depth increments (Fig. 2a). The $\Delta^{14}C$ of the soil TOC decreased with increasing soil depth at all three sites. In the uppermost 160 cm, the mean $\Delta^{14}C$ of the TOC decreased on average from -103 ‰ to -761‰ at the arid site, from 70‰ to -267 ‰ at the mediterranean site, and from -30 ± 35 ‰ to -469‰ at the humid site (Fig. 2a). The $\Delta^{14}C$ of the $CO_2$ respired by soil microorganisms also decreased with depth, but less than the $\Delta^{14}C$ of the soil TOC (Fig. 2b). In the uppermost 1 m of the soils, the $\Delta^{14}C$ of the respired $CO_2$-C

tended to be positive, while it was negative at depth > 1 m. The respired $CO_2$-C at a depth of 200–600 cm had a higher $\Delta^{14}C$, and thus was younger at the humid than at the mediterranean site (Fig. 2b). The $\Delta^{14}C$ of the $CO_2$ respired by soil microorganisms was higher than the $\Delta^{14}C$ of the TOC in the same depth increments in the soils of all three sites (except for two depth increments from the topsoil of the mediterranean site) (Fig. 2c), indicating that the respired C is younger than the soil TOC. The slope of the linear model that describes the $\Delta^{14}C$ of the respired C as a function of the $\Delta^{14}C$ of the TOC was

0.15 for the Mediterranean site, 0.23 for the arid site, and 0.26 for the humid site (Fig. 2c).

The $\delta^{13}C$ of the TOC increased at the humid site from a mean of -28.7 ± 0.2 ‰ in the organic layer to -23.2 ± 0.2 ‰ in the depth section 160–180 cm (Fig. 3a). At the mediterranean site, it increased from -27.2 ± 0.6 ‰ in the organic layer to -23.1 ± 0.3 ‰ at a depth of 40–60 cm, and did not change consistently with depth in the subsoil. At the arid site, the $\delta^{13}C$ of the TOC changed hardly in the whole soil profile (Fig. 3a). The $\delta^{13}C$ of the roots did not change substantially with depth (Fig. 3b). The

$\delta^{13}C$ of the roots increased in the order humid<mediterranean<arid in all depth increments (Fig. 3b), similar as the $\delta^{13}C$ of the TOC (Fig. 3a).

The concentration of microbial DNA increased in the order arid<mediterranean<humid across all depth increments (Fig. 4). In all soils, the DNA concentration decreased strongly with depth (Fig. 4). Similarly, the TOC concentration increased in the order arid<mediterranean<humid across all depth increments (Fig. 5). The TOC concentration decreased by 93 % at the arid,

and by 96 % at the mediterranean and humid site, respectively, in the uppermost 200 cm from top to bottom (Fig. 5). The microbial respiration rate in the soils of the humid and mediterranean site was higher than at the arid site in all depth sections (Fig. S1). The microbial respiration rates decreased by more than 99 % at all three sites in the uppermost 200 cm from top to bottom (Fig. S1).

## 4. Discussion

*4.1 Microorganisms live on young carbon, even several meters below the soil surface*

We found that microbial respiration in the soils of all three climate zones is fueled by modern C in the uppermost 1 meter of the soils and by relatively young C, as compared to the TOC, below 1 meter depth (Fig. 2a, b, c). Most C that is respired by





microorganisms in the upper part of the soil is likely directly derived from roots. This is consistent with the fact that annual dead root biomass inputs to soil make up about 10 and 34 % of the fine root biomass in forests and shrublands, respectively

(Gill and Jackson, 2000), while root exudates amount to 20–40% of the photosynthetically fixed C (Badri and Vivanco, 2009). Although roots are absent below 160 cm (arid site) and 350 cm (humid site) (Fig. 3b), the C respired by microorganisms in the deep soil is still relatively young (Fig. 2b, c), suggesting that microbial activity is mostly fueled by young dissolved organic carbon (DOC) that is rapidly transported downwards in the soil. Indeed, studies have shown that DOC fluxes from the top to the subsoil can be as high as 200 kg ha$^{-1}$ yr$^{-1}$ (Michalzik et al., 2001). Notably, the respired $CO_2$-C at a depth of 200–600 cm

tends to be younger at the humid than at the mediterranean site (Fig. 2b and c). As the humid site experiences higher precipitation and has a larger C content in the topsoil than the other two sites, it is likely that the transport of young C from the topsoil to the deep soil is the largest at this site. This possibility is supported by a meta-analysis that found a positive correlation between precipitation and DOC fluxes from the topsoil to the subsoil (Michalzik et al., 2001).

Our findings suggest that microorganisms throughout the soil, even several meters below the soil surface, metabolize organic

C that is relatively young, and substantially younger than the soil TOC. We interpret these findings to indicate preferential metabolism of organic C that is directly derived from roots or has been transported rapidly downwards from the topsoil to deeper soil horizons as DOC (Sanderman et al., 2008; Rumpel and Kögel-Knabner, 2011; Kaiser and Kalbitz, 2012; Marín-Spiotta and Hobley, 2022). These young organic matter inputs to soil are very likely not yet stabilized against microbial decomposition by sorption to mineral surface. In addition, they are likely more carbohydrate-, and thus energy-rich than

organic matter that has already been processed by the microbial community (Ni et al., 2020). Yet, it has to be considered that the soil was sieved before its incubation. Sieving destroys soil aggregates and potentially renders organic C available to microorganisms that was previously protected in microaggregates. Thus, it is likely that under *in situ* conditions, the $CO_2$-C that is respired by microorganisms is even younger (and thus has an even higher $\Delta^{14}$C) than in our incubation experiment.

Our findings are in accord with studies about topsoil, such as Trumbore (2000), who reported that whereas TOC in boreal,

temperate, and tropical forest topsoils was between 200 to 1200 years old, the $CO_2$-C respired by heterotrophic bacteria in these soil types was only 30, 8, and 3 years old, respectively. Since our study is the first to report the $\Delta^{14}$C of $CO_2$ respired by soil microorganisms below 1.0 m in non-permafrost soils, it sheds light on a hitherto unknown interaction in the critical zone. It shows that microbial processes in deep soil are closely connected to primary production aboveground. Owing to the replication of this study in three climate zones, our result that microbial activity in the deep soil is fueled mostly by young

organic matter may be generalizable across a large part of the terrestrial biosphere.

### *4.2 Strong decomposition of soil organic matter only occurs in the upper decimeters of the soils*

While soil organic matter moves downwards slowly, it interacts with the soil matrix and gets partly decomposed by microorganisms (Sanderman et al., 2008; Kaiser and Kalbitz, 2012). Microbial enzymes preferably decompose organic





compounds with a high proportion of the lighter $^{12}$C, which leads to enrichment of the heavier $^{13}$C isotope in soils (Ehleringer
et al., 2000; Wynn et al., 2006; Balesdent et al., 2018). In the Coastal Cordilleran soils, we found that strong decomposition of
organic matter that leads to $^{13}$C enrichment was restricted to the upper decimeters of the soils (Fig. 3a). This is likely because
organic matter in soils gets increasingly stabilized against microbial decomposition over time due to occlusion and sorption to
mineral surfaces, which renders the organic compounds spatially inaccessible (Kögel-Knabner et al., 2008; Schmidt et al.,
2011; Dungait et al., 2012). In addition, the proportion of carbohydrate-rich plant necromass decreases with depth, while the
proportion of carbohydrate-poor microbial necromass increases, which leads to a decrease in the energy-content of the soil
organic matter with increasing soil depth (Ni et al., 2020), and might be another reason why decomposition of TOC in the deep
soil is low. In addition, it needs to be considered that the change in the carbon stable isotope ratio might also be partly related
to the dilution of atmospheric $^{13}$C-$CO_2$ by $^{13}$C-depleted $CO_2$ derived from burning of fossil fuels (i.e. "Suess effect", Keeling
et al. 1979). However, it seems unlikely that this is the major reason for the observed change in the $\delta^{13}$C of the TOC given that
the decreases in $\delta^{13}$C with depth varies strongly among the three sites, and should be more similar if it were caused by a change
in the $\delta^{13}$C of atmospheric $CO_2$.

At the arid site, there is no enrichment of $^{13}$C in the soil TOC with increasing soil depth (Fig. 3a), indicating that little organic
matter processing by microorganisms occurs in this soil, probably because of the lack of moisture which hampers
decomposition (Moyano et al., 2013; Seuss et al., 2022). At the mediterranean and the humid sites, the soil TOC $\delta^{13}$C increases
with increasing depth down to 80 and 180 cm soil depth, respectively (Fig. 3a), suggesting that microbial processing of organic
matter occurred at greater depths at the humid site than at the mediterranean site, which is likely because of the higher soil
moisture at the humid site than at the mediterranean site.

### 4.3 The age of total soil organic carbon increases with soil depth and aridity

While the soil TOC content decreases with soil depth and aridity of the sites (Fig. 5), the age of the TOC increases (Fig. 2a),
suggesting that deeply weathered soils, particularly in an arid climate, can retain C from the atmosphere for long periods of
time. Our findings agree with a meta-analysis (Mathieu et al., 2015) that found that soils in an arid climate have a lower $^{14}$C
abundance in the topsoil and, particularly, in the deeper layers than soils in a humid climate. Consistent with our results, the
mean differences in $\Delta^{14}$C among dry and humid climate in that study amounted to 40 ‰ in the topsoils and 100 ‰ in the
subsoils, indicating that the soil TOC in arid areas is older than in humid areas, particularly in the subsoil (Mathieu et al.,
2015). This is partly because the microbial community in the subsoil feeds primarily not on TOC but on a different carbon
pool, derived from roots and young DOC that likely has a very short turnover time (see above). Another reason for why the
soil TOC at the arid site is particularly old is likely the low microbial biomass (Fig. 4) and activity (Fig. S1) caused by water
limitation and low TOC content at this site. This is supported by *in situ* measurements of the soil $CO_2$ concentration at the arid
site at different soil depths that detected low $CO_2$ concentrations throughout the year (Spohn and Holzheu, 2021).



### 4.4 Conclusions

Based on the unique measurements of microbially respired $^{14}C$-$CO_2$ down to a soil depth of 6 meters in different climate zones, this study reveals that microbial activity several meters below the soil surface is fueled by recently fixed C and that strong microbial decomposition of the soil TOC only occurs in the upper decimeters of the soils. Thus, in contrast to deep permafrost soils, in which microorganisms respire old C during large parts of the year, microorganisms in deep, non-permafrost soils feed on recent C inputs. Our results suggests that deeply-developed soils, not affected by permafrost, can restrain C from the atmosphere for climate-relevant periods of time because the microbial community in the subsoil mostly feeds on a different carbon pool, derived from roots and young DOC. Taken together, our results show that different layers of the Critical Zone are tightly connected, and that processes in the deep soil depend on primary production aboveground.

### Code Availability

Not applicable

### Data availability

All data will be made publically available once the manuscript is accepted for publication.

### Author contributions

MS and CS conceptualized the study, MS and AS conducted the field work, AS conducted the lab work, AS did the data analysis with input from MS and CS, MS wrote the manuscript, AS and CS contributed to the manuscript, MS acquired funding for the project.

### Competing interests

The authors declare that they have no conflict of interest.

### Acknowledgement

We are grateful to the Chilean National Park Service (CONAF) for providing access to the sample locations and on-site support for our research. We thank the whole Deep Earthshape team for collaboration during the field sampling campaigns. We thank Renate Krauß and Harald Isreal Suaznabar Olguin for their technical support in the laboratory. We acknowledge the Laboratory of Isotope Biogeochemistry at the University of Bayreuth for stable isotope measurements. We acknowledge the support from the $^{14}C$ Analysis Facility at the Max Planck Institute for Biogeochemistry in Jena, and thank Axel Steinhof and Manuel Rost for their technical support. The study was funded by the German Science Foundation (DFG) as part of the priority research program SPP-1803 "EarthShape: Earth Surface Shaping by Biota" (grant DFG SP1389/5-2).



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





**Tables**

**Table 1:** Properties of the three sites in the Coastal Cordillera in Chile, including mean annual temperature (MAT), mean annual precipitation (MAP), and the locations of the soil profiles.

| Climate zone | Site name | MAT [°C] | MAP [mm yr⁻¹] | Biome | Profile | Longitude | Latitude |
|---|---|---|---|---|---|---|---|
| Arid | Santa Gracia | 16.1 | 87 | Arid shrubland | 1 | -71.159439 | -29.759769 |
| | | | | | 2 | -71.160226 | -29.759037 |
| | | | | | 3 | -71.161234 | -29.759465 |
| Mediterranean | La Campana | 14.9 | 436 | Mediterranean sclerophyllous forest | 1 | -71.043710 | -33.028375 |
| | | | | | 2 | -71.041269 | -33.028585 |
| | | | | | 3 | -71.047170 | -33.028718 |
| Humid | Nahuelbuta | 14.1 | 1084 | Humid, temperate forest | 1 | -72.95065 | -37.79371 |
| | | | | | 2 | -72.95125 | -37.79017 |
| | | | | | 3 | -72.94868 | -37.79533 |
| | | | | | 4 | -72.95206 | -37.79517 |




**Figures**

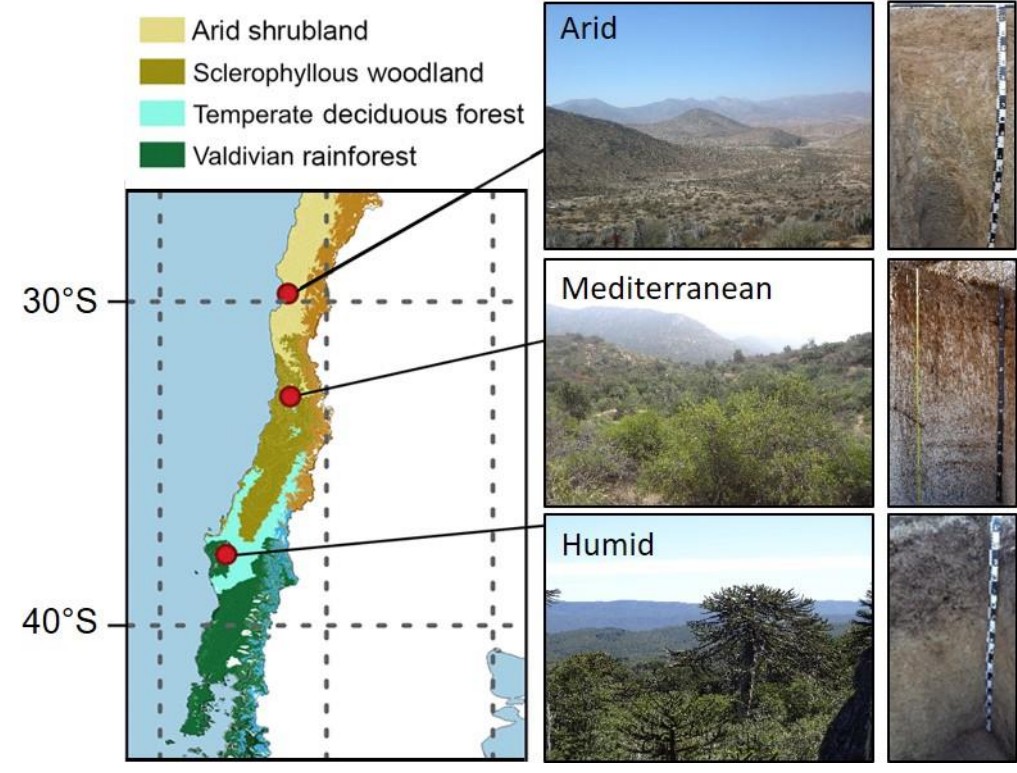

**Ble**

**Figure 1:** Location of the three study sites in the Coastal Cordillera of Chile together with a photo of the vegetation and soil at each of the sites. The map is taken from Werner et al. (2018) and was created based on Luebert and Pliscoff (2017).



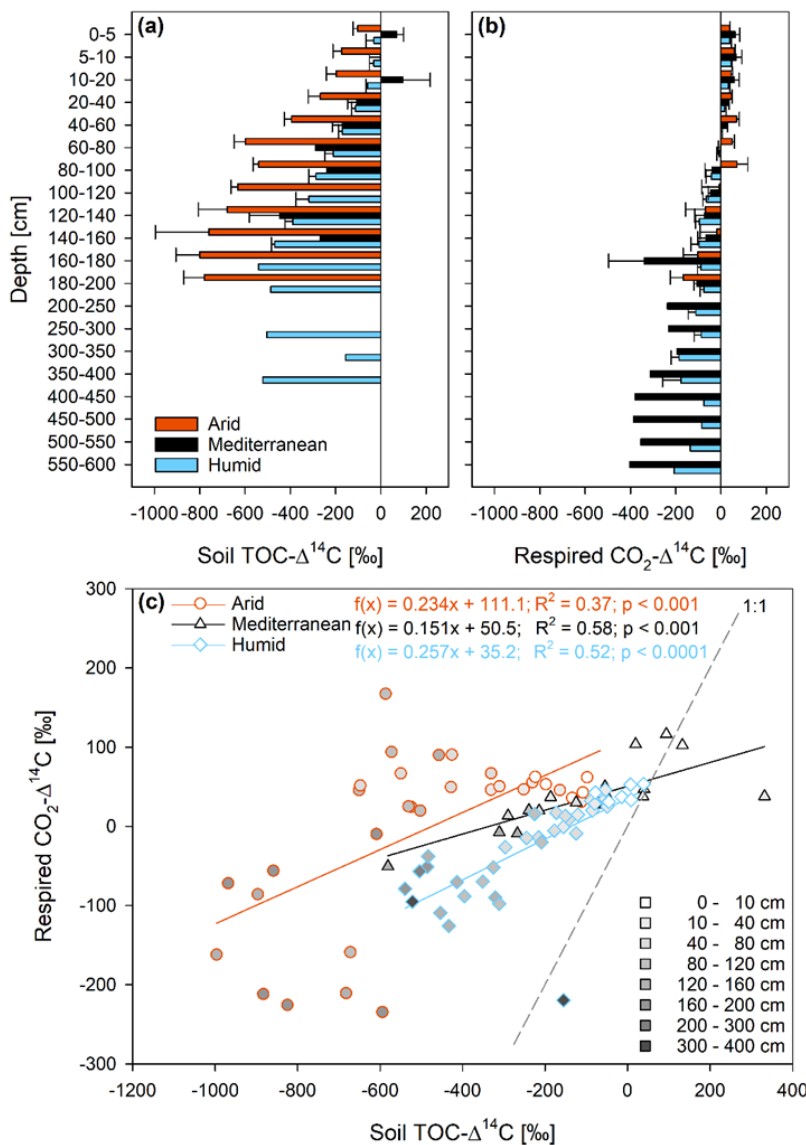

**Figure 2:** Radiocarbon ratio ($\Delta^{14}$C; mean ± standard error) of **(a)** total soil organic carbon (TOC) and **(b)** respired $CO_2$-C as well as **(c)** their relationship in soils at three sites (arid, mediterranean, humid) located along a precipitation gradient in the Coastal Cordillera of Chile (0-200 cm: n = 3, except for humid site with n = 4; > 200 cm: n = 1, except for humid site 200-400 cm with n = 2).

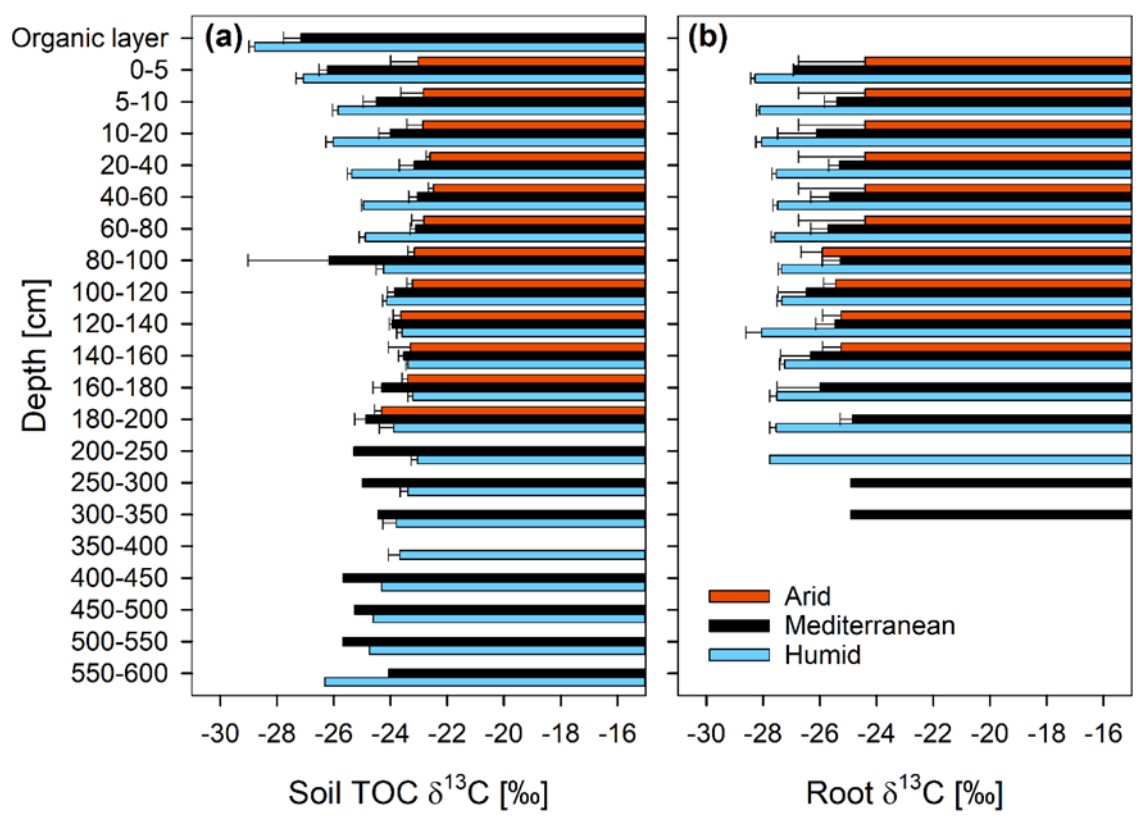

**Figure 3:** $^{13}$C ratio ($\delta^{13}$C; mean ± standard error) of **(a)** soil total organic carbon (TOC) and **(b)** roots at three sites (arid, mediterranean, humid) located along a precipitation gradient in the Coastal Cordillera of Chile (0-200 cm: n = 3, except for humid site with n = 4; 200-600 cm: n = 1, except for humid site 200-400 cm with n = 2).

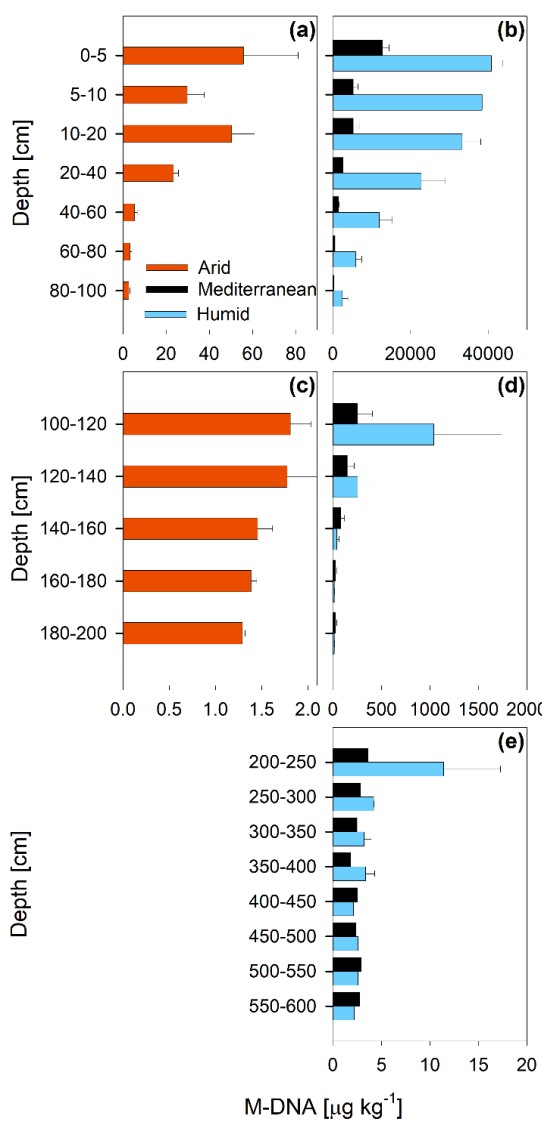

**Figure 4:** Microbial DNA content (M-DNA; mean ± standard error) of soil depth increments down to **(a, b)** 100 cm, **(c, d)** 200 cm, and **(e)** 600 cm depth at three sites (arid, mediterranean, humid) located along a precipitation gradient in the Coastal Cordillera of Chile (0-200 cm: n = 3, except for humid site with n = 4; > 200 cm: n = 1, except for humid site 200-400 cm with n = 2).


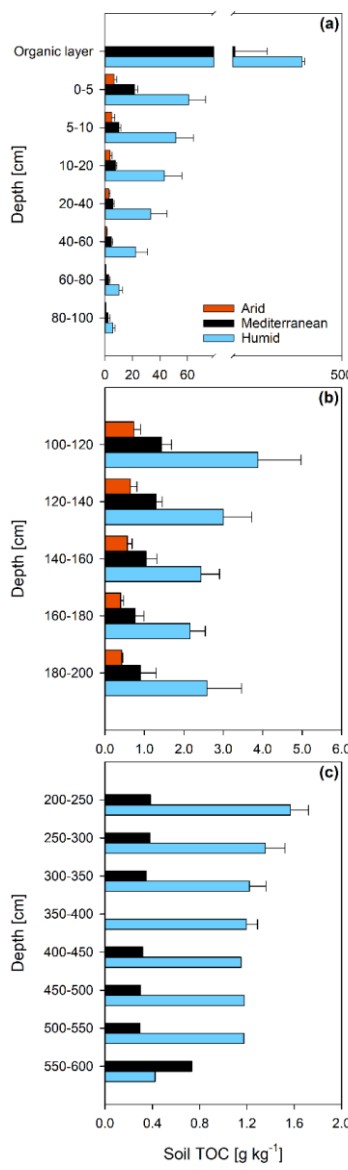

**Figure 5:** Total organic carbon (TOC) content (mean ± standard error) of soil depth increments down to **(a)** 100 cm, **(b)** 200 cm, and **(c)** 600 cm depth at three sites (arid, mediterranean, humid) located along a precipitation gradient in the Coastal Cordillera of Chile (0-200 cm: n = 3, except for humid site with n = 4; > 200 cm: n = 1, except for humid site 200-400 cm with n = 2).