# Peer review of "Recently fixed carbon fuels microbial activity several meters below the soil surface"

_Biogeosciences, 2022_

## Author Comment (AC1)

**Answer to the comments of reviewer 1**

In this paper, the authors present results from a unique study comparing soil carbon stock and 14C values to soil respiration fluxes and 14C values during soil incubation of soils from 3 sites with distinct climate to up to 6 meters depth. What is truly unique about this study is the depth to which the authors sampled and incubated soils. I also find the depth increments to be impresively fine, adding to the value of these data (especially for modeling). The authors found that even at several m depth, microbial respiration during incubations is fueled by relatively younger C. While this finding is not unique, the comparison of sites across a climate gradient and the depth to which this study sampled in the soil profile is quite unique.

The paper is reasonably well written, but could be improved with some editing and revisions to the figures. These are relatively minor issues. More concerning, however, is the lack of statistical analyses or descriptions to support the authors' interpretations and conclusions. This maniscript needs some description of the statistical analyses used to provide the reported errors (are they standard deviation or standard error? something else?) and at minimum some simple statistical tests to look at differences between the sites and the 14C of TOC vs respired CO2-C. This could be simple: A t-test to test the statistical significance of the TOC vs CO2 could be done easily by calculated the difference and testing if it is different from 0 - this could be a paired test so the same site/depth are compared to one another.

We thank the reviewer very much for the positive and constructive review of our manuscript. We added the results of the statistical analyses to all figures, and we now explain the statistical analyses that we conducted in a newly added section (section 2.8) in Material and Methods. In addition, we improved the manuscript according to the detailed comments (see below).

More detailed comments are below:

The title is great - I love that the main finding is right there! Thanks!

L 19 "higher than of the soil" is missing a "that" or something similar We added "that".

L 23 "strong microbial decomposition" - what is meant by "strong"? Perhaps a different word is better. We replaced "strong" by "most".

L 24 "which is likely due to stabilization" - I do not follow this logic. Do we know that decomposition stops at depth because of stabilization? I do not know that that we know that for sure. Perhaps "posssibly" instead of "likely" or "partly" or be more specific about what you mean exactly. We added "that leads to enrichment of $^{13}$C" in this sentence and replaced "likely" by "possibly", as suggested.

L 30 "topsoils" are not generally thought of as the top meter (often this is the surface or A horizon, maybe the top 10 cm or 20 cm or even 30 cm, but certainly not the top meter!) Consider just saying in the top meter here and elsewhere, "top meter" is not much longer than "topsoil". We replaced "topsoil" by "uppermost meter of soils" here, and checked the correct use of the term topsoil throughout the manuscript.

L 35 the following paragraph is confusing to me - do you mean in the field? Incubations? Can you be more specific? We added "in incubations" in line 36 and "measured in the field" in line 42.

L 43 you need references to back up the statement that the total CO2 is largely composed of CO2 respired by roots. And what do you mean by largely? Half? Can you be more specific? What did these studies find? We replaced "is" by "can be" and we added a reference.

L 44 This paragraph would be a good place to clarify that you did laboratory incubations. It is not clear that you used incubations until the methods on L 92. We added "in incubations" in line 52.

L 137 This looks like a total DNA extraction with no clean up to remove plant or animal DNA - is this true? If so you should not call it microbial DNA. If you did remove plant and animal DNA please explain this more clearly. I think you can still use total DNA as a proxy for microbial DNA and biomass but you should be clear that this is what you have. Yes, in order to clarify this we removed "microbial".

L 149 use "flushed" or "scrubbed" instead of "rinsed" done as suggested.

L 152-153 I don't follow - samples were collected at multiple time points over 11 or 8 months and checked for CO2 concentration? It sounded like it was collected for 14C as it was written but the results seem to have only one time point for each site. I don't think you need to provide so much detail here - you can just say CO2 concentrations were monitored and you collected the samples for 14C when there was enough or when you cutoff the incubation (at 11 months?) Please clarify. There is also detailed description of the respiration rate sampling earlier, but those data are not presented - why is that? We removed the details and simply say that the $CO_2$ concentration was monitored. The respiration rate is presented in a figure in the Supplement.

L 167 what is the reported analytical uncertainty? were soils pretreated for carbonates? were there carbonates (especially at the arid site?) The soils did not contain carbonates (as clearly stated in line 86. We now added the following sentence to the description of the AMS measurement "The analytical uncertainty is 2 ‰".

L 172 I know what you mean, but the phrasing is awkward. You could say "...model to the data, but there was not enough data to constrain such a model" that said you could make assumptions or do a 2 compartment model with the respiration data, if you wanted to. (I don't think you do though!) We changed the sentence as suggested.

L175 - there is no description of statistics for the results! How were trends and differences assessed? Are these differences statistically significant? What are the reported uncertainties? Standard deviation or error or something else? Where is the respiration rate data? The respiration data was and is in the Supplement. All figure present(ed) standard errors and this is/was clearly explained in the figure captions. We now added results of statistical tests to all figures (see our answer to the first point raised by the reviewer).

L 204 Why do you say that most C respired by microorganisms is likely directly derived by roots? Can you be more specific? You might consider refering at some point to Phillips et al 2013: https://bg.copernicus.org/articles/10/7999/2013/ We added "This is further supported by Philips et al (2013) who measured $^{14}C$-$CO_2$ in a temperate hardwood forest, and found that rhizodeposition was an important driver of microbial respiration".

L 223 add citations (you could use Phillips et al 2013 but there are likely others) We added the suggested citation.

L 227 What "interaction" do you refer to? Can you be more specific? This sentence is very awkwardly worded, please revise. We revised the sentence.

L 233 This paragraph is difficult to follow - is it about decomposition or about 13C? It seems to go back and forth but not clearly explain how the two are connected. We improved several sentences in this paragraph.

L 237 What do you mean by "this"? Stabilization would not cause enrichment in 13C. We replaced "this" by "organic matter decomposition is likely restricted to the topsoil".

L 240 I do not follow how this is connected to 13C or the rest of the paragraph, it feels disconnected from the rest. We shortened and changed this sentence.

L 244-6 it would only be similar if C was cycling the same way, it seems like C cycles more slowly at the arid site and you would expect to see a smaller Suess effect there, which you do see. I would just cut this last sentence and reword the first part of the previous sentence to something like, "it is possible this is partly a result of the dilution of atmospheric 13C....." Done as suggested.

L 253 Earlier in the paper suggests that 13C of roots were used for this interpretation too, but it looks like no? Are those data helpful here? Were there carbonates anywhere (it looks like maybe not, but it would be helpful if you could clarify this). No, there were no carbonates (we had stated this clearly in the description of the soils).

L 250 There is no discussion about the differences in vegetation causing some of the differences between sites. You should at least consider these differences since there is definitely a difference in vegetation cover across sites. We added the following sentence "In addition, the higher decomposition of organic matter at greater depth at the humid site compared to the other sites might also be related to the trees at this site that seem to cause a larger root density in deeper soil horizons than the vegetation at the other sites, and thus likely stimulate microbial activity in the subsoil."

L 254 can you add some more citations? Maybe https://doi.org/10.1007/s10533-020-00725-z Done as suggested

L 260 by "a different carbon pool" do you mean "a portion of the TOC pool"? Yes. We revised this sentence.

L 263 don't you think this is mostly because of water? these sites are so dry! Yes, that is what we say ("water limitation").

L 270 this comparison to permafrost seems very out of place here. You could develop this comparison by adding more context to the Intro and to the Discussion (with citations). If you think this important it could make the paper more interesting. We refer to permafrost soils in the second paragraph of the Introduction. We now also added a comparison to permafrost soils in the Discussion.

L 273 what do you mean by "tightly connected"? Can you be more specific? It seems like the deeper soil is connected to the shallow soil, but are the shallow soils really connected to the deeper soils? We specify this in the same sentence ("that processes in the deep soil depend on carbon that recently entered the ecosystem through $CO_2$ fixation").

L 277 where will the data be? the paper has been accepted as a discussion paper, where can the data be found? We are currently in the process of publishing all data in the International Soil Radiocarbon Database, and the doi will be available very soon.

Figure 2 for figure c, you could add a 1:1 line to show where the points would fall if respired C = TOC (since the axis are not the same) Figure 2c already included a 1:1 line. For better recognition, we now added the 1:1 line in the legend.

Figure 3 It doesn't seem like the axis need to go to -16 delta 13C. Can you plot these across a smaller range to make it easier to see the differences? Maybe -22 to -30 permil? Done as suggested

Figure 4 It is difficult to compare the panels arranged this way - maybe you could plott all 3 sites together with an inset or break in the axis to help show the differences? Something more like figure 5? Done as suggested

---

## Author Comment (AC2)

**Answer to the comments of reviewer 2**

In this preprint the authors use carbon (C) isotopes ($^{14}C$ and $^{13}C$) to determine the source and age of respired soil carbon in three sites along a climatic gradient in the Coastal Cordillera of Chile. They find that respired CO2-C was of more recent origin than soil organic C and that C in deeper soils was older than surface C. They then use these results along with total DNA extracted from the soils to conclude that microbial decomposition is primarily of new carbon, rather than old in these soils. I found this manuscript to be an important contribution to our understanding of the origin and age of microbially-processed soil carbon and generally an interesting well-written piece. My chief concerns are to do with the authors' use of total DNA as equivalent to microbial DNA, and therefor a proxy for microbial activity (Line 137), in some of the authors statistical methods relating respired total $^{14}C$ to respired $^{14}C$ (Figure 2c), and the interpretations regarding the influence of primary productivity (Line 228, conclusions, and final line in abstract), which I have detailed below.

We thank the reviewer very much for the positive and constructive review of our manuscript. Our detailed answers can be found below.

Specific comments below:

Line 49-50: It would be nice to include an explanation for why the arid site was dug less deeply than the other two, perhaps in the methods section 2.2. We added the following sentence in section 2.2. "The soils profiles were chosen to be less deep at the arid site because the soils at this site are less deeply developed than at the other two sites (Bernhard et al., 2018)."

Line 73: Do the authors have any estimation of the differences in isotopes that might be due to the differences in sampling year and how that compares to the ranges seen in their results? This is particularly important for the $^{13}C$-isotope results. I could see those as being particularly sensitive to differences in temperature and moisture conditions between the two sampling years. The samples were collected in the same month (March) of two consecutive years. Therefore, we have no reason to believe that the difference in sampling year affects our ($^{13}C$-isotope) results. The $^{14}C$ ratios are corrected for sampling year (as stated in the method description).

Line 114: For methodological transparency, the authors should include a brief explanation about how they determined the pre-incubation period for each environment. The pre-incubation varied between 3 (arid site) to 4 (mediterranean and humid sites) days for practical reasons. Given that the experiment lasted for six weeks, and that the soils of the sites differ in very many respects this difference should be of very minor importance.

Line 137: Rather than referring to this measurement as microbial DNA, the authors should refer to it as total DNA. It likely not only includes DNA from microbial sources (e.g. microbial eukaryotes, fungi, and prokaryotes), but also DNA from non-microbial sources such as plant roots or soil arthropods. Although proportionally the non-microbial DNA is likely to be low in comparison, it is likely to vary among soil types (based on the amount of vegetation and moisture of the soil) and depth. Therefore it likely that the soils not only contain to contain different proportions of microbial:non-microbial DNA in different soils but also at different depths. The authors therefore should be cautious in their interpretation of total DNA as a proxy for microbial activity and should adjust their discussion accordingly. One relatively

easy experimental way around this, would be to use qPCR to quantify the abundance of 16S and ITS gene copies in each soil sample's DNA. These numbers would be a more accurate quantification of microbial abundance at least, even though not all organisms possessing those genes are likely to have been active during incubation. We changed microbial DNA to total DNA throughout the manuscript, as also recommended by the first reviewer.

Section 2.7 – not having much disciplinary expertise in this area, I defer to others who do in evaluation of the methods. However, I appreciate the explanation in the last paragraph about accurate estimation of ages, which is helpful for non-experts in $^{14}$C dating such as myself. It is also helpful that the authors imply that the carbon they observe likely is derived from the last 1000 years, and would be further helpful for their non-expert audience to know where the estimate of 1000 years comes from. Furthermore in their discussion, the authors describe a similar study in permafrost that was able to estimate ages using $^{14}$C, if there is any way to get at least a range of ages from this data to compare to that study, I would find it very useful in interpreting the results. We are aware that much confusion exists in the literature regarding the reporting of ages from soil radiocarbon data. It is true that many previous studies have reported a conventional or a calibrated radiocarbon age, but only recently there has been more awareness in the soil radiocarbon community about this issue and lack of consistency in reporting ages (e.g. Trombone et al. 2016, Sierra et al. 2017, https://doi.org/10.1111/gcb.13556). The explanation we give at the end of section 2.7 is the main reason we do not provide an age estimate, even though some previous studies have done so. We do not want to continue reproducing this practice of reporting radiocarbon ages when we know that in an open system such as a soil all the organic matter is a mix of carbon from a wide spectrum of ages (e.g. Chanca et al. 2022, https://doi.org/10.1029/2021JG006673). We mention that we would need data of carbon inputs from at least the last 1000 years to accurately model this system in the method description. The mentioned 1000 years is only a rough estimate and by no means an accurate estimate of how old the carbon in these soils may be.

Figure 2c – The $R^2$ value of the arid linear model is quite low compared to the mediterranean and humid sites. I wonder if a simple linear model is even appropriate for this relationship as it seems that depth, along with other co-correlates are at play. In fact, for the humid site as well, the mean and variance appear to be related with higher mean values having less variance at more shallow sites which violates the assumption of equal variance of residuals which means that their estimated slopes may be incorrect. The authors should address this low fit in the text and potentially may find some helpful solutions in this guide: https://academic.macewan.ca/burok/Stat378/notes/remedies.pdf  The linear models provide a reasonable representation of the relationship. The main purpose of these regression lines is to underline that the datapoints (almost) all plot below the 1:1 line.

Line 228, conclusions, and final line in abstract: I'm not sure the authors have made a very strong argument that processes of the soil are highly dependent on primary productivity aboveground. Although the results do indicate that newer carbon is being respired, an alternative explanation could be that the newer carbon is simply being re-cycled among the microbial community as the community turns over. The incubations are not directly measuring the influence of primary productivity since they are plant-free incubations and there aren't measurements for primary productivity at each site. I recommend a more conservative interpretation. We replaced "primary production aboveground" by "recent carbon inputs from plants" and "carbon that recently entered the ecosystem through $CO_2$ fixation." in abstract and conclusion. Even though we cannot exclude that the C has already

cycled through the microbial biomass before being respired, we know that it has recently been fixed from the atmosphere.

Section 4.2: The authors may also want to consider the process of priming in their interpretation of this observation (see for example Bernard et al. 2022 for a nice review on the subject - https://besjournals.onlinelibrary.wiley.com/doi/full/10.1111/1365-2435.14038). We added the following sentence "Further, it could be that recently fixed carbon that enters the soil in the upper decimeters also leads to priming (Bernard et al., 2022)."

Technical corrections:

Line 98 – Typo, I believe "weight" should be "weighed" Corrected

Line 118 – typo in the specification of volume of gas: there are extra spaces between the number and the units. Additionally the units of liters should be abbreviated with a capital "L" rather than lowercase "l". Corrected.

---

## Author Response (AR2)

Sveriges lantbruksuniversitet
Swedish University of Agricultural Sciences

**Department of Soil and Environment**
Dr. Marie Spohn
Professor of Biogeochemistry of Forest Soils

18. January 2023

To
Biogeosciences
Editor-in-chief Sara Vicca

Dear Sara,

Thank you very much for your review of our manuscript and for your letter.

We now conducted Two-Way ANOVA followed by Tukey's posthoc test. The results are indicated in the figures and in a new table (Table S2).

We look forward to hearing from on you on the manuscript.

Kind regards,

Marie Spohn

Mailing address: marie.spohn@slu.se
Street address: Lennart Hjemls väg 9
Phone: +46 18 67 10 00 (switchboard)
Mobile: +46 730321569

---

## Author Response (AR3)

**Responses to the editor**

We thank the editor for the comments, which helped us to further improve the manuscript.

1. You used a two-way ANOVA to test for differences between sites and soil depths and their interaction effects, but the revised text does not mention any interaction effects. Was this an oversight? I would like to suggest to briefly mention the interaction effect in the Results section.

We added the following information in line 187 "yet the interaction effect between site and soil depth was not statistically significant (Table S2)", and the following sentence in lines 198-199: "The  $\delta^{13}C$  of the TOC differed significantly between sites, and also the interaction effect of site and soil depth was statistically significant (Fig. 3a and Table S2)."

2. ANOVA-type analyses do not require normal distribution of the raw data, but of the residuals of the ANOVA model. This is a common misconception - see <a href="https://pubs.er.usgs.gov/publication/5224239">https://pubs.er.usgs.gov/publication/5224239</a>. While non-normality in response or explanatory variables can indeed make it more likely that model assumptions are not met (e.g., homoscedascity), the patterns in your model residuals should guide you in choosing the type of analysis or data transformation. Please revise accordingly.

We had tested the residuals of the model and not the raw data for normality and homoscedasticity. For a more accurate description of our statistical analyses, we reworded the sentence in the method section in the following way. *"For this purpose, the residuals of the Two-Way ANOVA model were tested for normality using the Shapiro-Wilk test and for variance homogeneity using the Levene's test."*

3. Please provide a reasoning/motivation for doing some analyses per soil depth increment (section 2.8): "Significant differences between radiocarbon data (14C) of the soil total organic carbon (TOC) and respired CO2-C for individual soil depth increments per site were analyzed using the Welch's t-test (in case of normality) or the Wilcoxon rank sum test (in case of violation of normality)."

All variables were tested using Two-Way ANOVA. We now added the following sentence in section 2.8 to provide a motivation for the additional analysis. "In addition, we tested whether TOC and microbialderived CO2-C from the same soil depth increment differ significantly in their 14C signature in specific soil depth increments since differences might potentially only occur in specific depth ranges." The details of the analysis are described in the next sentence in the manuscript.